# The angiopoietin-like protein ANGPTL4 catalyzes unfolding of the hydrolase domain in lipoprotein lipase and the endothelial membrane protein GPIHBP1 counteracts this unfolding

Simon Mysling[1,2,3], Kristian Kølby Kristensen[1,2], Mikael Larsson[4], Oleg Kovrov[5], André Bensadouen[6], Thomas JD Jørgensen[3], Gunilla Olivecrona[5], Stephen G Young[4,7], Michael Ploug[1,2]*

[1]Finsen Laboratory, Rigshospitalet, Copenhagen, Denmark; [2]Biotech Research and Innovation Centre, University of Copenhagen, Copenhagen, Denmark; [3]Department of Biochemistry and Molecular Biology, University of Southern Denmark, Odense, Denmark; [4]Department of Medicine, University of California, Los Angeles, Los Angeles, United States; [5]Department of Medical Biosciences, Umeå University, Umeå, Sweden; [6]Division of Nutritional Science, Cornell University, Ithaca, United States; [7]Department of Human Genetics, University of California, Los Angeles, Los Angeles, United States

**Abstract** Lipoprotein lipase (LPL) undergoes spontaneous inactivation *via* global unfolding and this unfolding is prevented by GPIHBP1 (*Mysling et al., 2016*). We now show: (1) that ANGPTL4 inactivates LPL by catalyzing the unfolding of its hydrolase domain; (2) that binding to GPIHBP1 renders LPL largely refractory to this inhibition; and (3) that both the LU domain and the intrinsically disordered acidic domain of GPIHBP1 are required for this protective effect. Genetic studies have found that a common polymorphic variant in ANGPTL4 results in lower plasma triglyceride levels. We now report: (1) that this ANGPTL4 variant is less efficient in catalyzing the unfolding of LPL; and (2) that its Glu-to-Lys substitution destabilizes its N-terminal $\alpha$-helix. Our work elucidates the molecular basis for regulation of LPL activity by ANGPTL4, highlights the physiological relevance of the inherent instability of LPL, and sheds light on the molecular defects in a clinically relevant variant of ANGPTL4.

*For correspondence: m-ploug@finsenlab.dk

## Introduction

The intravascular hydrolysis of triglycerides in triglyceride-rich lipoproteins (TRLs) by lipoprotein lipase (LPL) is an efficient mechanism for delivering lipid nutrients to vital tissues (*e.g.*, heart, skeletal muscle, adipose tissue). LPL is secreted into the interstitial spaces by myocytes and adipocytes but is then shuttled to the capillary lumen by an endothelial transporter, glycosylphosphatidylinositol-anchored high density lipoprotein–binding protein 1 (GPIHBP1) (*Fong et al., 2016*). This endothelial cell LPL transporter is also required for the margination of TRLs along capillaries (*Goulbourne et al., 2014*). Disruption of LPL transport by either genetic ablation of GPIHBP1 in mice (*Beigneux et al., 2007*) or by loss-of-function missense mutations in humans leads to sustained elevations in plasma triglyceride levels—a condition known as familial chylomicronemia (*Ariza et al., 2016*; *Beigneux et al., 2015*, *2009b*; *Olivecrona et al., 2010*; *Plengpanich et al., 2014*).

Appropriate distribution of lipid nutrients to tissues is, however, not only controlled by the availability of LPL and GPIHBP1, but is also tightly regulated by endogenous protein inhibitors of LPL activity (angiopoietin-like (ANGPTL) proteins 3, 4, and 8). Reciprocal regulation of ANGPTL4 expression in brown and white adipocytes with prolonged cold exposure results in high LPL activity in brown adipocytes and ensures the delivery of lipid nutrients for adaptive thermogenesis (*Dijk et al., 2015*). In the fed state, this balance is reversed, resulting in high LPL activity in white adipose tissue and allowing replenishment of lipid stores in white adipose tissue. The relevance of these inhibitors to LPL-mediated TRL processing in vivo is illustrated by lower plasma triglyceride levels with pharmacological inhibition or genetic ablation of ANGPTL3 or ANGPTL4 (*Desai et al., 2007*; *Gusarova et al., 2015*; *Köster et al., 2005*). Additional evidence for their physiological importance are provided by genome-wide association studies (GWAS), which showed that a polymorphic variant of ANGPTL4, E40K (E15K in the mature protein) is associated with mild hypotriglyceridemia and reduced risk of coronary artery disease (*Dewey et al., 2016*; *Helgadottir et al., 2016*; *Stitziel et al., 2016*). Throughout this paper, we will use amino acid numbering starting at the amino terminus of the mature proteins.

Although genetic and physiologic evidence emphasizes the importance of ANGPTL-mediated regulation of LPL-activity, our understanding of the biochemical and structural basis for this inhibitory activity remains incomplete. An early report suggested that the N-terminal coiled-coil domain of ANGPTL4 catalyzed the conversion of LPL homodimers to inactive LPL monomers (*Sukonina et al., 2006*). However, this conclusion has been challenged by a report suggesting that ANGPTL4 inhibits LPL by a reversible and noncompetitive mechanism (*Lafferty et al., 2013*). Thus far, this controversy is unsettled.

In the current study, we take advantage of hydrogen–deuterium exchange mass spectrometry (HDX-MS) to probe the structural changes in LPL that accompany both spontaneous decay of LPL activity and ANGPTL-mediated inhibition of LPL activity. Our data show that ANGPTL4 does in fact catalyze the irreversible inactivation of LPL and does so by promoting unfolding of LPL´s hydrolase domain. Notably, we also document that the polymorphic variant ANGPTL4$^{E15K}$ has a lower capacity to inhibit LPL activity and lower capacity to catalyze the unfolding of LPL´s hydrolase domain, compared with wild-type ANGPTL4. This reduced activity is associated with destabilization and cooperative unfolding of the N-terminal α-helix in ANGPTL4, a region of ANGPTL4 that had previously been implicated in LPL inhibition (*Lee et al., 2009*; *Yau et al., 2009*).

## Results

### Spontaneous unfolding of LPL is catalyzed by ANGPTL4

We previously showed that LPL irreversibly loses catalytic activity by undergoing a time- and temperature-dependent global unfolding of its serine hydrolase domain (*Mysling et al., 2016*). In contrast, the C-terminal lipid-binding domain of LPL remains stably folded. Since LPL activity in vivo is regulated in an organ-specific manner by ANGPTL 3, 4, and 8 (*Dijk et al., 2015*; *Dijk and Kersten, 2016*; *Wang et al., 2015*, *2013a*), we speculated that these proteins might catalyze the unfolding of LPL´s hydrolase domain. To explore this possibility, we incubated 10 µM bovine LPL, either alone or in the presence of 1 µM purified ANGPTL4$^{1–159}$, at 25°C in 8.8 mM Na$_2$HPO$_4$, 0.7 mM HCOOH, and 150 mM NaCl, pH 6.9. The time-dependent unfolding of LPL´s hydrolase domain was probed by changes in hydrogen–deuterium exchange profiles with short (10 s) exposures to D$_2$O. Deuterium exchange was quenched by lowering the pH, and the deuterium uptake into different regions of LPL was assessed by on-line pepsin digestion and mass spectrometry (*Mysling et al., 2016*). This approach provides an imprint of regional changes in protein stability within LPL under variable conditions. Mass spectrometry revealed that 10 µM LPL undergoes a slow spontaneous transition into an unfolded state during the selected incubation time interval (0–30 min), with 29% of the LPL being unfolded after 30 min at 25°C (*Figure 1*). The amount of LPL unfolding was quantified by the gradual progression towards a bimodal signature in the isotope envelope for the 'diagnostic' peptic peptide 131–165 from LPL´s hydrolase domain – *Figure 1A* (*Mysling et al., 2016*). Quantitative assessment of this distribution reveals that 7.0 ± 0.7% of LPL was in the unfolded state after 5 min. Adding 1 µM ANGPTL4$^{1–159}$ to 10 µM LPL greatly accelerated the rate of LPL unfolding, with 60.3 ± 2.3% of the LPL being unfolded after 5 min (*Figure 1B*). Even though the amount of ANGPTL4$^{1–159}$ added was

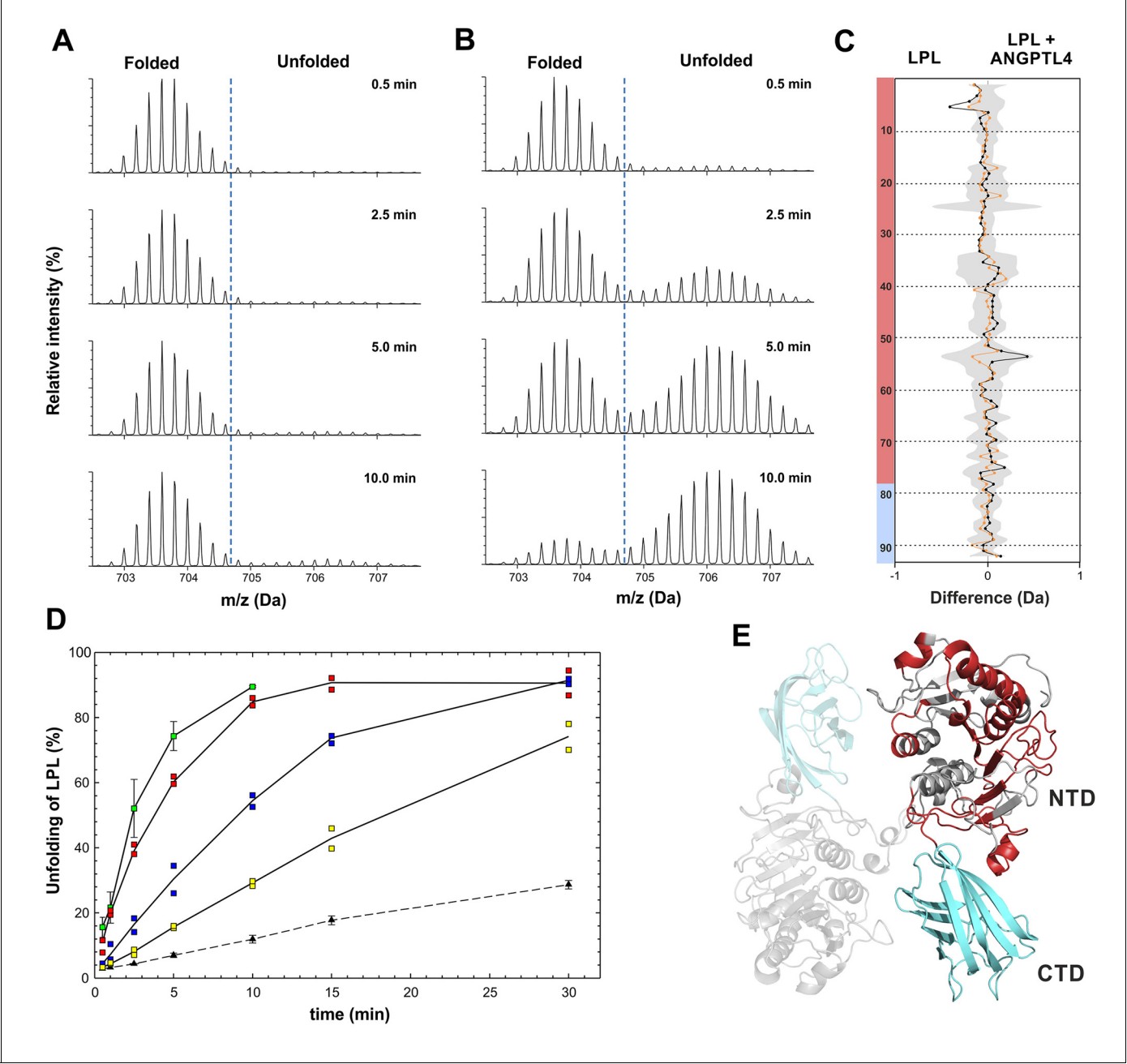

**Figure 1.** Kinetics of spontaneous and ANGPTL4-mediated unfolding of LPL′s serine hydrolase domain. Unfolding of the hydrolase domain in LPL is revealed by the bimodality in the isotope envelopes for the peptide 131–165, which contains two residues from LPL′s catalytic triad (Ser[134] and Asp[158]). Time-resolved isotope signatures of this diagnostic peptide are shown for 10 μM LPL incubated alone (*panel* A) or in the presence of 1 μM ANGPTL4[1–159] (*panel* B). The butterfly plot in *panel* C compares the spontaneous and ANGPTL4-catalyzed unfolding of LPL at conditions where 10–15% (*orange curve*) or 15–20% (*black curve*) bimodality for peptide 131–165 is observed. The deuterium uptakes were calculated from the average mass of the bimodal isotope envelopes. The shaded gray area corresponds to the largest standard deviation in the data sets recorded for each peptide (triplicates). Transparent red and cyan colors on the left assign these peptides to LPL′s N-terminal hydrolase domain (NTD) or its C-terminal lipid-binding domain (CTD), respectively. The identities of the 92 peptic LPL peptides are found elsewhere (*Mysling et al., 2016*). *Panel* D shows the time-dependent unfolding of 10 μM LPL either alone (▲) or in the presence of ANGPTL4[1–159] (□) at 2, 1, 0.5, and 0.25 μM, as indicated by *green, red, blue*, and *yellow* coloring, respectively. This value is calculated as the ratio between the bimodal isotope envelopes of the folded and unfolded proteins. *Panel* E shows a homology model of LPL with the proposed head-to-tail homodimer formation indicated by the semitransparent LPL subunit (*Kobayashi et al., 2002*; *Wang et al., 2013b*). The red color highlights the regions in LPL showing bimodality of isotope envelopes (*i.e.,* unfolding) by HDX-MS.

The following figure supplements are available for figure 1:

*Figure 1 continued on next page*

*Figure 1 continued*

**Figure supplement 1.** Estimating the catalytic efficacy of ANGPTL4$^{1–159}$ on LPL unfolding.

**Figure supplement 2.** Dose- and time-dependent inhibition of LPL catalytic activity by ANGPTL4$^{1–159}$.

**Figure supplement 3.** Oligomerization status of purified ANGPTL4$^{1–159}$.

**Figure supplement 4.** ANGPTL4-mediated inhibition of LPL at 4°C —impact of deoxycholate (DOC).

substoichiometric relative to LPL (a molar ratio of 1:10), ANGPTL4 was responsible for unfolding >50% of LPL molecules in 5 min. This noticeable amount of LPL unfolding by substoichiometric amounts of ANGPTL4 constitutes a strong argument in favor of the catalytic nature of ANGPTL4-mediated LPL inactivation (*Sukonina et al., 2006*). Because the kinetic profiles for LPL unfolding with 1 µM ANGPTL4$^{1–159}$ suggested that substrate depletion occurred (*Figure 1D*), we also tested the ability of lower amounts of ANGPTL4 (0.5 µM, 0.25 µM) to unfold LPL. Those studies provided clear evidence for a dose-dependent inactivation of LPL by ANGPTL4 (*Figure 1D*). The lowest ANGPTL4 concentration tested resulted in 74 ± 6% LPL unfolding after 30 min compared to a spontaneous unfolding of 28.6 ± 1.3% in the buffer lacking ANGPTL4. Even though ANGPTL4$^{1–159}$ was present at a molar ratio of 1:40 (relative to LPL), it nevertheless resulted in the unfolding of 50% of LPL molecules. From early time point measurements, where substrate depletion was negligible, we estimate that the unfolding efficacy of our ANGPTL4$^{1–159}$ preparation was 1.6 ± 0.2 molecules of LPL per ANGPTL4 molecule per min at 25°C (*Figure 1—figure supplement 1*). When identical incubations of LPL and ANGPTL4 were analyzed for lipolytic activity with [$^3$H]triolein, we found a similar time- and dose-dependent inhibition of LPL catalytic activity by ANGPTL4$^{1–159}$ (*Figure 1—figure supplement 2*).

To compare the conformational changes associated with spontaneous and ANGPTL4-catalyzed inactivation of LPL, we examined the deuterium uptake values for all 92 peptic peptides recovered from LPL. To standardize the alignment of these data, we selected conditions that led to uniform and relatively low levels of bimodality in the 'diagnostic' peptide 131–165. By compiling all data recorded for spontaneous and ANGPTL4-catalyzed unfolding, we generated two bins of data with 10–15% or 15–20% bimodality in the isotope envelopes for peptide 131–165. In comparing the deuterium uptake across all LPL peptides, we observed no differences in the pattern of deuterium uptake into LPL molecules undergoing spontaneous or ANGPTL4-catalyzed unfolding (*Figure 1C*). Of note, all peptides showing bimodality were confined to the serine hydrolase domain of LPL (*Figure 1E*); none were in the C-terminal lipid-binding domain. Based on these data, we conclude that different means of LPL inactivation are mirrored by similar conformational endpoints whether they occur spontaneously or are catalyzed by ANGPTL4.

Several reports show that ANGPTL4 is prone to various degrees of oligomerization *via* intermolecular disulfide bonds involving Cys$^{51}$ and Cys$^{55}$ in ANGPTL4´s N-terminal coiled-coil domain (*Ge et al., 2004a*, *2004b*; *Yin et al., 2009*). Our ANGPTL4$^{1–159}$ preparation also contained disulfide-linked oligomers, although they were predominantly in the form of dimers and comprised no more than 10% of the protein preparation (*Figure 1—figure supplement 3A*). In monomeric ANGPTL4, we find that Cys$^{51}$ and Cys$^{55}$ are connected by an intramolecular disulfide bond, as demonstrated by masses of 3440.64 and 3442.64 Da for peptides corresponding to residues 45–77 in ANGPTL4$^{1–159}$ before and after reduction with tris-(2-carboxyethyl) phosphine (TCEP) (*Figure 1—figure supplement 3B and C*). To exclude the possibility that a minor fraction of disulfide-linked ANGPTL4 oligomers are responsible for all of ANGPTL4´s catalytic unfolding activity, we reduced and alkylated the disulfide-bonds in ANGPTL4 and then compared the LPL unfolding capacity with ANGPTL4 harboring intact disulfide bonds. As shown in *Figure 1—figure supplement 3D*, the catalytic activity of 1 µM ANGPTL4$^{1–159}$ was similar regardless of the integrity of the disulfide bond. In conclusion, a covalent oligomer of ANGPTL4$^{1–159}$ is not essential for its catalytic LPL-unfolding activity in vitro. Nevertheless, we cannot formally exclude the possibility that a non-covalent assembly of ANGPTL4$^{1–159}$ oligomers participates in LPL unfolding.

# GPIHBP1 binding mitigates the ANGPTL4-catalyzed inactivation of LPL

The proper expression of GPIHBP1 is critical for shuttling LPL to the capillary lumen, where margination and lipolytic processing of TRLs occurs (*Beigneux et al., 2009a*; *Davies et al., 2010*; *Fong et al., 2016*; *Goulbourne et al., 2014*). Recently, we uncovered a new functionality of GPIHBP1—stabilizing LPL activity by limiting the spontaneous unfolding rate of LPL's hydrolase domain (*Mysling et al., 2016*). These findings prompted us to ask whether the binding of GPIHBP1 to LPL would also protect LPL from ANGPTL4-catalyzed unfolding. To explore this possibility, we first incubated 10 μM LPL alone or with 30 μM GPIHBP1 for 2 min at 25°C; we then added 2 μM ANGPTL4$^{1-159}$ and incubated the mixture for 10 min. In the absence of GPIHBP1, we observed 87 ± 2% unfolding of LPL *versus* only 8 ± 2% unfolding in the presence of GPIHBP1$^{1-131}$ (*Figure 2A*). Comparable results were obtained with spontaneous unfolding of LPL (*Figure 2B*). Both the LU domain and the acidic domain of GPIHBP1 (*Figure 2C*) were required for protecting LPL against ANGPTL4; neither 30 μM GPIHBP1$^{34-131}$ nor 30 μM GPIHBP1$^{1-33}$ substantially inhibited ANGPTL4-catalyzed unfolding of LPL (79 ± 7% and 69 ± 12% unfolding, respectively). A slightly different scenario emerged for spontaneous unfolding of LPL; the acidic domain of GPIHBP1 (GPIHBP1$^{1-33}$) was nearly as effective as full-length GPIHBP1$^{1-131}$ in protecting LPL from unfolding (*Figure 2B*). To analyze this difference in more detail, we tested the potency of a mixture of two GPIHBP1 domains (30 μM GPIHBP1$^{1-33}$ and 30 μM GPIHBP1$^{34-131}$) on spontaneous and ANGPTL4-catalyzed LPL unfolding. The GPIHBP1$^{1-33}$/GPIHBP1$^{34-131}$ mixture was less effective than full-length GPIHBP1 in protecting LPL from ANGPTL4-mediated unfolding (*Figure 2A*). In conclusion, it would appear that GPIHBP1's acidic domain needs to be tethered to the LU domain to achieve GPIHBP1's full protective effect against ANGPTL4-catalyzed LPL unfolding (*Figure 2A*). A covalent association of the two GPIHBP1

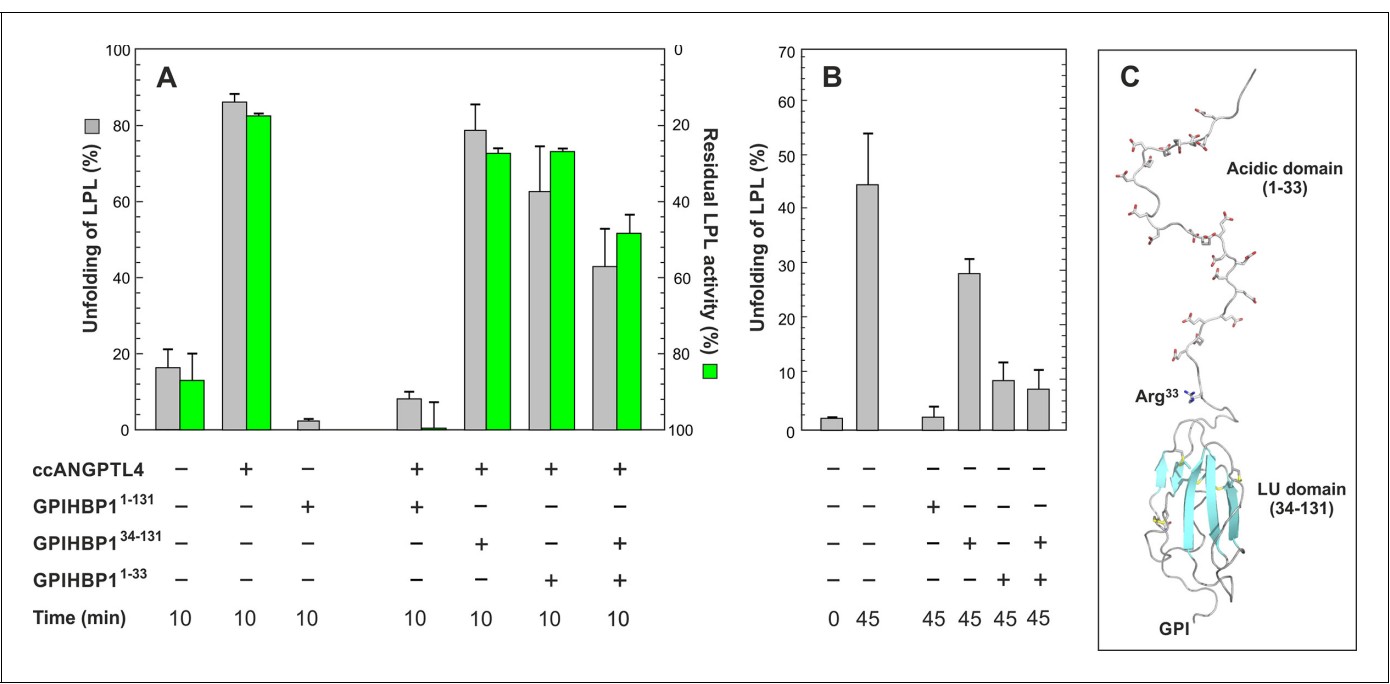

**Figure 2.** GPIHBP1 binding counteracts ANGPTL4-catalyzed unfolding of LPL. *Panel A* shows the degree of unfolding of 10 μM LPL incubated for 10 min at 25°C with 2 μM ANGPTL4$^{1-159}$ alone or in the presence of 30 μM GPIHBP1$^{1-131}$, 30 μM GPIHBP1$^{34-131}$, 30 μM GPIHBP1$^{1-33}$, or a mixture of 30 μM GPIHBP1$^{34-131}$ and 30 μM GPIHBP1$^{1-33}$ (*gray bars*). The loss in lipolytic LPL activity on samples treated identically is shown by *green bars*. These activity data were normalized relative to the initial LPL activity assuming a spontaneous decay of 15% during the experiment. *Panel B* shows the degree of spontaneous unfolding of 10 μM LPL before and after a 45 min incubation at 25°C either alone or in the presence of 10 μM GPIHBP1 in the same combinations used in *panel A*. The relative unfolding of LPL is calculated from the bimodal distribution of the isotope envelopes for peptide 131–165. *Panel C* shows a molecular model for GPIHBP1 depicting the intrinsically disordered acidic domain (GPIHBP1$^{1-33}$) and the ordered LU domain (GPIHBP1$^{34-131}$). The acidic residues in the N-terminal domain are highlighted by sticks as is the Arg$^{33}$, which is exposed and prone to proteolysis (*Mysling et al., 2016*). The attachment site for the membrane anchor is indicated (GPI).

domains is less important for reducing the spontaneous unfolding of LPL (*Figure 2B*). In these studies, we also analyzed catalytic activity of LPL. The protection provided by the different GPIHBP1 proteins against ANGPTL4-mediated LPL unfolding was accompanied by preserved LPL catalytic activity towards [³H]triolein substrate hydrolysis (*Figure 2A*).

To further interrogate the mode of action of ANGPTL4-catalyzed LPL inhibition, we added 300 nM intact GPIHBP1 [12-fold above the $K_D$ for the GPIHBP1•LPL interaction (*Mysling et al., 2016*)] to LPL (15 nM) that had been partially inactivated—either by incubating the enzyme at 21°C (spontaneous inactivation) or by incubating the enzyme with 7.5 nM ANGPTL4 (catalyzed inactivation). As shown in *Figure 3*, adding GPIHBP1 to the partially inactivated LPL protected the remaining LPL from inactivation, but neither the spontaneously inactivated LPL nor the ANGPTL4-inactivated LPL was resurrected by GPIHBP1. These findings align with the HDX-MS data (*Figure 1C*) showing that both the spontaneous and ANGPTL4-mediated LPL inactivation pathways result in very similar (if not identical) conformational changes in LPL. Our studies further substantiates that ANGPTL4 catalyzes LPL unfolding/inactivation in an irreversible manner rather than acting as a noncompetitive and reversible inhibitor as previously suggested (*Lafferty et al., 2013*). If ANGPTL4-mediated inactivation of LPL were to be reversible, we would have expected to find reactivation of LPL during the incubation with GPIHBP1.

## Unfolding properties of ANGPTL$^{E15K}$

Genetic studies have revealed that a polymorphic variant in ANGPTL4 (ANGPTL4$^{E15K}$, with a ~3% prevalence in Caucasians) is associated with lower plasma triglyceride levels and reduced risk of coronary artery disease (*Dewey et al., 2016*; *Helgadottir et al., 2016*; *Stitziel et al., 2016*). We expressed and purified the coiled-coil domain of ANGPTL4$^{E15K}$. The kinetic profile of LPL unfolding with ANGPTL4$^{E15K}$ was more complex and included a time-dependent decline in LPL-unfolding activity (using conditions in which substrate depletion was not rate-limiting) (*Figure 4A*). We were able to take advantage of the linear phases of the progress curves for LPL unfolding to estimate the unfolding efficacy of ANGPTL4$^{1-159/E15K}$. The ANGPTL4 variant resulted in the unfolding of 0.6 ± 0.1 molecules LPL per ANGPTL4 molecule per min (*Figure 4B and C*), which is only 40% of the activity of wild-type ANGPTL4 (*Figure 4C*). We obtained consistent results when measuring LPL activity; the LPL incubated with ANGPTL4$^{1-159/E15K}$ retained 40–50% more catalytic activity than LPL incubated with wild-type ANGPTL4 (*Figure 4D*). The LPL unfolding curves (*Figure 4A and B*) suggested that ANGPTL4$^{1-159/E15K}$ lost activity more rapidly than wild-type ANGPTL4 during the LPL incubation.

## Protein dynamics of ANGPTL4

Given the differences in the kinetics of LPL-unfolding catalyzed by ANGPTL4$^{wt}$ and ANGPTL4$^{E15K}$, we embarked on determining the intrinsic protein dynamics of the two ANGPTL proteins with HDX-MS. We incubated 3 μM ANGPTL4$^{1-159}$ in 10 mM $Na_2HPO_4$, 150 mM NaCl (pH 7.1) containing 70% (v/v) $D_2O$ for 10, 100, or 1000 s at 25°C. We recovered 41 peptides from ANGPTL4$^{1-159}$ after on-line pepsin digestion (corresponding to 95.6% sequence coverage) (*Figure 5—figure supplement 1A and B*). As illustrated by the heat maps for deuterium uptake, wild-type ANGPTL4$^{1-159}$ contains two regions that are less dynamic (residues 19–43 and 85–111). Both sequences are predicted by PSIPRED to form α-helices (*Figure 5A*). Of the two α-helices, the N-terminal α-helix of ANGPTL4 is less stable (*i.e.*, has a greater deuterium uptake). The isotope envelopes reveal that the deuterium uptake in this particular region occurs *via* mixed EX1 and EX2 kinetics signifying some degree of cooperative unfolding (*Weis et al., 2006*).

A comparison of the kinetics for deuterium uptake into ANGPTL4$^{1-159/E15K}$ and ANGPTL4$^{1-159/wt}$ revealed a conspicuous difference: peptides covering residues 19–43 exchanged more rapidly in ANGPTL4$^{1-159/E15K}$ (*Figure 5A and C*). The increased deuterium incorporation occurs predominantly via faster EX1 kinetics reflecting an increased propensity for cooperative unfolding of that region. Given that this region coincides with the predicted N-terminal α-helix in ANGPTL4, it is conceivable that the Glu-to-Lys substitution in the E15K variant compromises protein stability by reducing the α-helix propensity of the adjacent C-terminal sequence. Our findings are likely biological relevant, providing a plausible molecular explanation for the impaired LPL inactivation and lower plasma triglyceride levels in carriers of the E15K polymorphism.

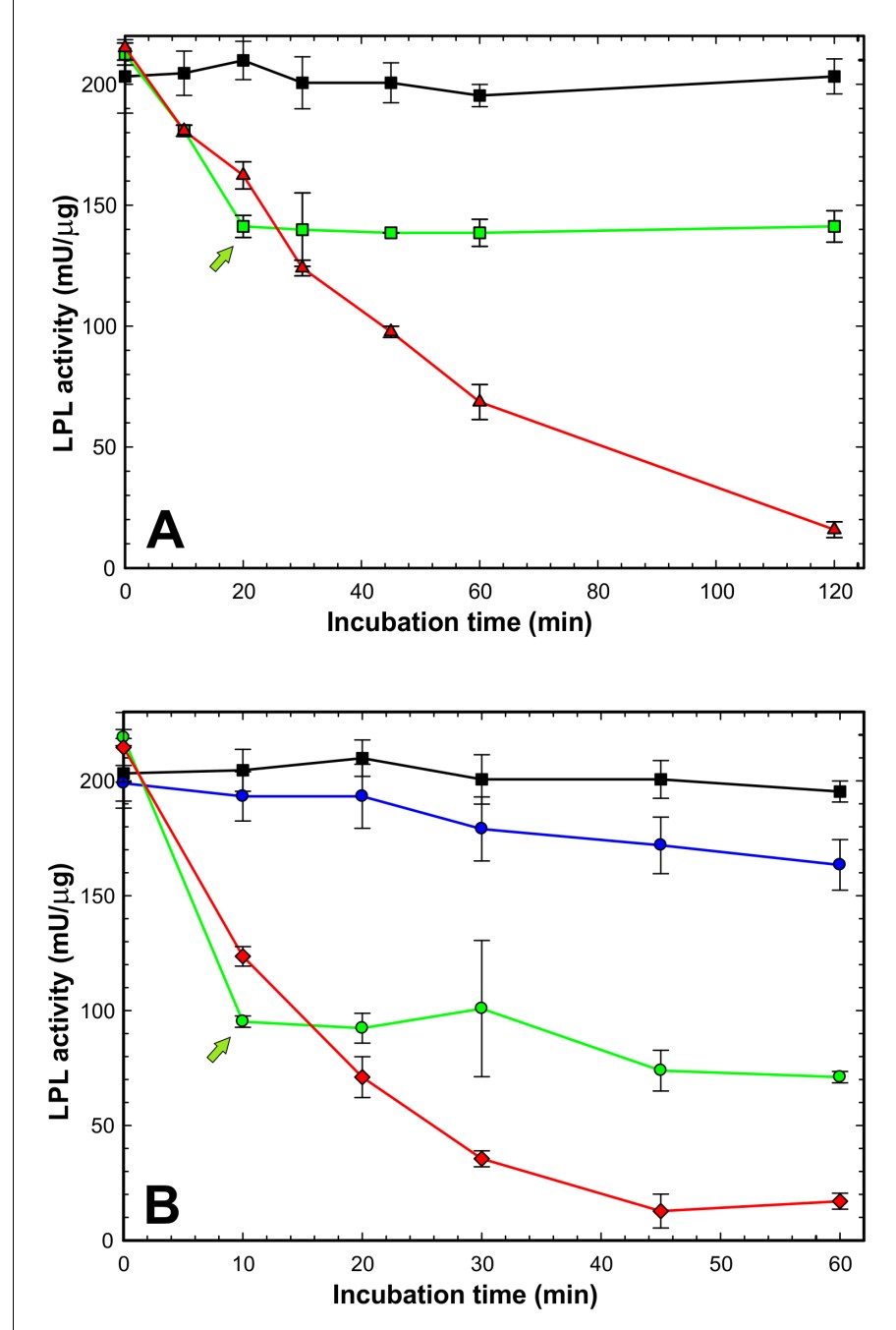

**Figure 3.** GPIHBP1 cannot reverse loss of LPL activity after spontaneous or ANGPTL4-catalyzed inactivation. *Panel* A shows time-dependent loss of LPL catalytic activity after incubating LPL at 21°C alone or in the presence of GPIHBP1. The progressive loss in catalytic activity of LPL (15 nM) during the incubation is shown by *red triangles*; the protection provided by a pre-incubation with GPIHBP1[1–131] (300 nM) is shown by *black squares*. Adding GPIHBP1[1–131] (300 nM) to LPL that was incubated alone for 20 min (*green arrow*) prevented any further loss of LPL (*green squares*). *Panel* B shows inactivation of LPL catalytic activity by 7.5 nM ANGPTL4[1–159] (*red diamonds*). The inhibition of LPL activity by ANGPTL4 was markedly reduced by the presence of 300 nM GPIHBP1[1–131] (*blue circles*); however, a small amount of ANGPTL4-mediated inhibition probably still persisted during the incubation with GPIHBP1, given that LPL catalytic activity was greater when LPL was incubated with GPIHBP1 in the absence of ANGPTL4[1–159] (*black squares*). When 300 nM GPIHBP1[1–131] was added to LPL 10 min after the initiation of ANGPTL4-mediated LPL inactivation (*green arrow*), there was no further loss of LPL activity (*green circles*). However, the loss of LPL activity that had occurred during the first 10 min of incubation was not reversed (*i.e.* there

*Figure 3 continued on next page*

*Figure 3 continued*
was no resurrection of ANGPTL4-inactivated LPL). LPL activity measurements were performed in triplicated; data shown ± SD.

## Unfolding catalyzed by intact ANGPTL3

Sequence alignments of the different ANGPTL proteins implicated in triglyceride metabolism (ANGPTL3, ANGPTL4, ANGPTL8) reveal strong sequence conservation in the N-terminal region that is destabilized in ANGPTL4$^{E15K}$ (*Figure 6A*). Interestingly, this region constitutes a part of the epitopes on ANGPTL3 or ANGPTL4 for monoclonal antibodies that mitigate LPL inactivation in vivo (*Lee et al., 2009*). In view of these observations, we tested whether purified ANGPTL3 might also serve as a catalyst for LPL unfolding. We found that ANGPTL3 also accelerates the unfolding of the catalytic triad peptide leading to inactivation of LPL, but at reduced efficacy compared with equimolar amounts of ANGPTL4 (*Figure 6B and C*). The ANGPTL3-catalyzed inactivation of LPL is abrogated by GPIHBP1 (*Figure 6C*).

## Discussion

The intravascular hydrolysis of triglycerides by LPL is the central event in plasma lipoprotein metabolism, providing fatty acids that are used as fuel in striated muscle and brown adipose tissue or stored in the form of triglycerides in white adipose tissue. Normally, LPL activity is tightly regulated, facilitating the delivery of lipid nutrients according to the metabolic demand of tissues and the physiologic state of the organism (*e.g.*, fasting, cold exposure, exercise). One of the most important mechanisms for regulating LPL activity relies on temporal and spatial repression by the inhibitors ANGPTL3, ANGPTL4, and ANGPTL8. The physiological importance of these inhibitors has come into focus as a result of genome-wide association studies and pharmacological and genetic studies in animal models (*Desai et al., 2007*; *Dewey et al., 2016*; *Gusarova et al., 2015*; *Helgadottir et al., 2016*; *Wang et al., 2013a*). However, uncertainty has persisted regarding the molecular mechanism(s) by which these inhibitors modulate LPL activity.

In the current study, we used HDX-MS analyses to study the inhibition of LPL by ANGPTL4. Our studies showed that ANGPTL4 induces a dose- and time-dependent unfolding of LPL´s hydrolase domain, and that these structural changes are accompanied by a parallel reduction in LPL's capacity to hydrolyze a lipid emulsion containing [$^3$H]triolein. Importantly, the unfolding of LPL´s hydrolase domain occurred with substoichiometric amounts of ANGPTL4. Combined, our data infer that ANGPTL4 acts by accelerating the spontaneous unfolding of LPL (*Mysling et al., 2016*). Our data are closely aligned with and extend the notion that ANGPTL4 acts as an 'unfolding molecular chaperone' (*Sukonina et al., 2006*). They are however not compatible with the view that ANGPTL4 act as a reversible and non-competitive inhibitor of LPL activity (*Lafferty et al., 2013*). The studies that led to the latter model involved incubating LPL in the presence of 1–5 mM deoxycholate (DOC), an anionic detergent that is known to bind to and stabilize LPL activity (*Bengtsson and Olivecrona, 1979*). When we replicated those experiments in the absence of DOC, we found a time-dependent inactivation of LPL activity consistent with an irreversible mechanism (*Figure 1—figure supplement 4*). We suspect that DOC changes the intrinsic dynamics of LPL's hydrolase domain, preventing it from visiting unstable conformations. In the absence of DOC, ANGPTL4 may bind transiently to LPL, stabilize a folding intermediate that 'guides' LPL towards irreversible unfolding, and ultimately cause permanent LPL inactivation. Whether this intermediate is an integrated part of the LPL homodimer or it represents a dissociated monomer is currently unclear. Importantly, our model for ANGPTL-catalyzed unfolding of LPL involves an ATP-independent mechanism unlike those generally employed by intracellular unfoldases (*Prakash and Matouschek, 2004*). Nevertheless, our model is not completely unprecedented as it bears some resemblance to the oligomerization of $\alpha_1$-antitrypsin that is catalyzed by a monoclonal antibody raised against its pathogenic (E342K) Z-variant (*Irving et al., 2015*).

Several studies have shown that ANGPTL4 undergoes a variable degree of oligomerization at or close to the cell surface *via* the formation of intermolecular disulfide bonds between Cys$^{51}$ and Cys$^{55}$ in different coiled-coil domains—and that the covalent oligomerization is biologically significant

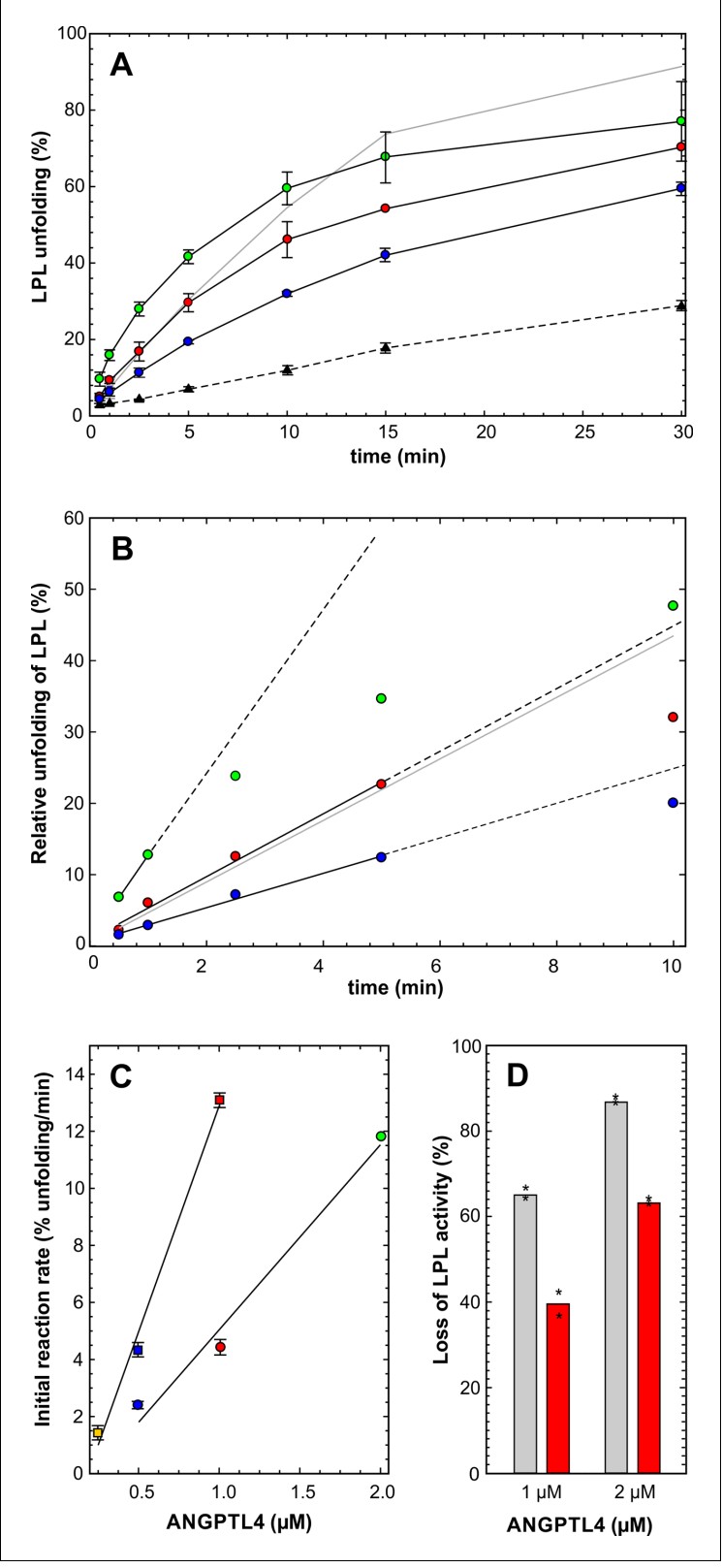

**Figure 4.** Impact of ANGPTL4$^{E15K}$ on LPL unfolding. *Panel* A shows the time-dependent unfolding of 10 µM LPL in the presence of 2, 1, and 0.5 µM ANGPTL4$^{1–159/E15K}$, highlighted by black solid lines and the *green, red* and *blue* circles, respectively. For comparison the unfolding mediated by 0.5 µM ANGPTL4$^{1–159/wt}$ is shown by the gray solid line. The hatched line represents spontaneous LPL unfolding. The corresponding data are shown in *panel* B after

*Figure 4 continued on next page*

*Figure 4 continued*

subtraction of spontaneous unfolding. The solid black lines show the data used to estimate the unfolding rates and their hatched extension illustrates nonlinearity at longer incubation times. The unfolding rates of ANGPTL4$^{1-}$$^{159/E15K}$ (*circles*) and ANGPTL4$^{1-159/wt}$ (*squares*) are represented by the slopes of the lines in *panel* C. The difference in the inhibitory efficacies between ANGPTL4$^{1-159/wt}$ (*gray bars*) and ANGPTL4$^{1-159/E15K}$ (*red bars*) on LPL-mediated hydrolysis of [$^3$H]triolein is shown in *panel* D (asterisks represent the individual measuring points).

(*Ge et al., 2004a*, *2004b*; *Makoveichuk et al., 2012*; *Yin et al., 2009*). In the current study, we show that the redox states of Cys$^{51}$ and Cys$^{55}$ do not influence ANGPTL4-mediated catalysis of LPL unfolding. Consistent with our findings, an independent study showed that the LPL inhibitory potential of ANGPTL4 in vitro was not altered when Cys$^{51}$ and Cys$^{55}$ were changed to serine (*Shan et al., 2009*). The ability of ANGPTL4—lacking disulfide bonds in the coiled-coil domain—to inactivate LPL is consistent with recent studies showing that ANGPTL4 can inactivate LPL intracellularly and reduce its secretion from cells (*Dijk et al., 2016*). In this compartment ANGPTL4 does not form covalent oligomers (*Makoveichuk et al., 2012*). Rather than being important for the catalytic properties of ANGPTL4, the covalent oligomerization could serve to increase its biological stability and ensure a proper extracellular partitioning of the coiled-coil domain of ANGPTL4 in vivo (*Yin et al., 2009*).

LPL-mediated hydrolysis of triglycerides occurs predominantly at discrete foci along the capillary bed where lipoproteins are sequestered by LPL•GPIHBP1 complexes on the endothelial cell membrane (*Fong et al., 2016*; *Goulbourne et al., 2014*). Given that the majority of LPL in the capillaries is bound to GPIHBP1, as judged with immunohistochemistry, it becomes pertinent to evaluate the inhibitory property of ANGPTL4 in that context. In an earlier study (*Sonnenburg et al., 2009*), soluble GPIHBP1, but not heparin, protected LPL's esterase activity from ANGPTL4-catalyzed inactivation. In our current study, we also found that GPIHBP1 protected LPL´s catalytic activity, and this protection was achieved by blocking ANGPTL4-catalyzed unfolding of LPL´s hydrolase domain (*Figure 2*). Noteworthy, full-length GPIHBP1 was required for robust protection of LPL from ANGPTL4; a mixture of GPIHBP1's LU and acidic domains provided only a modest protection. In view of those observations, we propose a molecular model in which the LU domain of GPIHBP1 secures a tight binding to the carboxyl-terminal domain of LPL. This assembly allows an optimal positioning of GPIHBP1's intrinsically disordered acidic domain close to the unstable catalytic domain of LPL protecting it from ANGPTL4-catalyzed unfolding.

One implication of our findings is that only GPIHBP1-bound LPL would be protected from ANGPTL4 in vivo. We predict that this selectivity is relevant for regulating lipid metabolism in several different compartments.

First, in the secretory pathway of adipocytes, where GPIHBP1 is absent, we suspect that LPL would be more susceptible to ANGPTL4-mediated unfolding. Accordingly, recent studies of LPL production by mouse primary adipocytes and white adipose tissue explants indicate that ANGPTL4 interacts with LPL in the *trans*-Golgi network, targeting it for lysosomal degradation (*Dijk et al., 2016*). The endoplasmic reticulum has an efficient quality control system that targets thermodynamically unstable proteins for disposal via ERAD (*Ellgaard and Helenius, 2003*). Emerging evidence suggests that an equivalent quality control system may also exist in the *trans*-Golgi network. In *Saccharomyces cerevisiae*, the trafficking receptor Vps10p serves as a protein-folding sensor diverting misfolded proteins to vacuolar disposal (*Hong et al., 1996*). One of the Vps10p domain–containing homologs in mammals, the sorting receptor SorLA, is known for its role in the intracellular trafficking of amyloid precursor proteins and for its association with neurodegenerative diseases (*Andersen et al., 2016*). Interestingly, SorLA is also expressed by adipocytes and myocytes and targets newly synthesized LPL for degradation in lysosomes (*Klinger et al., 2011*). One possibility—which obviously needs experimental testing—is that ANGPTL4-catalyzed *unfolding* of LPL in the *trans*-Golgi network primes an anterograde sorting of SorLA-LPL complexes towards the endolysosomal degradation pathway. Such a mechanism might account for the increased LPL secretion from *Angptl4*$^{-/-}$ adipocytes compared to wild-type adipocytes (*Dijk et al., 2016*). In keeping with the latter idea, we speculate that human adipocytes that express ANGPTL4 E15K polymorphism would secrete elevated levels of LPL.

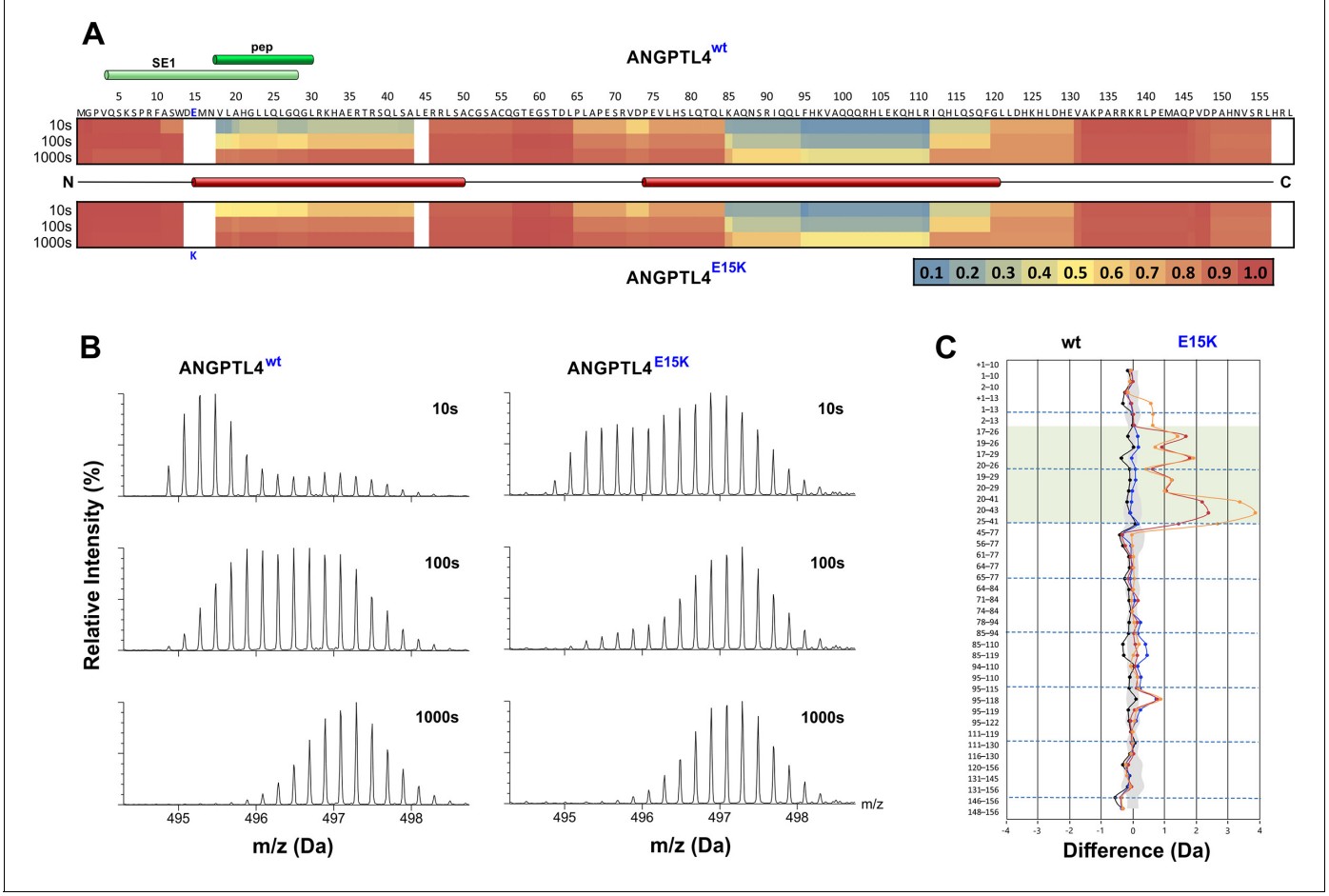

**Figure 5.** Protein dynamics of wild-type ANGPTL4 and ANGPTL4 containing the E15K polymorphism. Purified ANGPTL4$^{1-159}$ preparations (wt and E15K) were diluted to 10 μM with D$_2$O-phosphate buffer at pH 7.1, and hydrogen–deuterium exchange was monitored for 10, 100, and 1000 s at 25°C. *Panel* A provides a heat map of the deuterium uptake values (relative to a fully exchanged control) for peptic peptides from wt and E15K ANGPTL4$^{1-159}$ as assessed by HDX-MS. The relative deuterium uptake values for these peptides are plotted on the primary protein sequence [ranging from *blue* (no deuterium uptake) to *red* (full deuterium uptake)]. The position of the specific epitope 1 (SE1) defined by the neutralizing mAb 14D12 is shown by the *light green* cylinder (*Desai et al., 2007*; *Lee et al., 2009*), and the sequence representing an active synthetic peptide is shown by the *dark green* cylinder (*Yau et al., 2009*). The secondary structure of ANGPTL4 wt was predicted by PSIPRED (*Jones, 1999*) and the positions of the two α-helices are shown by *red* cylinders. The position of E15K polymorphism is shown by the *blue* letter. *Panel* B shows the isotope envelopes recorded for peptide 20–41 of ANGPTL4 wt and E15K; it shows progressive deuterium uptake as a function of labeling time. *Panel* C visualizes the differences in the dynamics of wt and E15K ANGPTL4$^{1-159}$ with a butterfly plot, showing differential deuterium uptake for the two proteins recorded after 10 s (*orange*), 100 s (*red*), and 1000 s (*blue*). The differential uptake for the full-deuterium exchange controls is shown for comparison (*black*). Peptides including residues 18–30 are highlighted by the transparent *green* box. The shaded *gray* area corresponds to the largest standard deviation in the data sets recorded for each peptide (triplicates). In panel A, note that the primary sequence starts with a methionine for the bacterial expression, but this is not included in the peptide numbering. In panel C, the presence of the methionine in a peptic peptide is denoted as +1.

The following figure supplement is available for figure 5:

**Figure supplement 1.** Peptide list for pepsin-treated ANGPTL4$^{1-159}$.

Second, freshly secreted LPL—loosely associated with heparan-sulfate proteoglycans (HSPG) in the interstitial spaces—would also be prone to inhibition by ANGPTL4 (*Olivecrona, 2016*). It has been suggested that this compartment represents the primary target site for ANGPTL4-catalyzed inactivation of LPL (*Nilsson et al., 2012*). In this setting, translocation of LPL from the HSPG-bound reservoir to a GPIHBP1-bound state at the basolateral surface of endothelial cells would alleviate the inhibitor repression exerted by ANGPTL4. We propose that inhibition of LPL within the interstitial

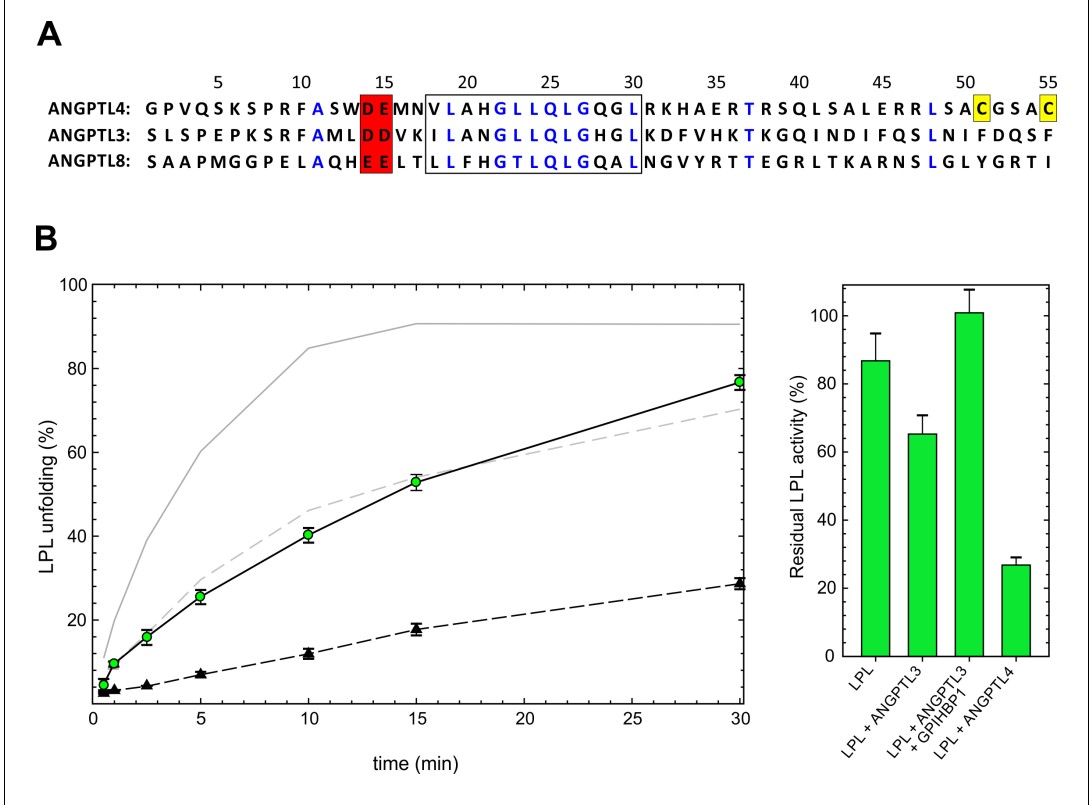

**Figure 6.** ANGPTL3-catalyzed unfolding of LPL. *Panel* **A** shows a sequence alignment of the first 55 residues of fully processed human ANGPTL3, ANGPTL4, and ANGPTL8. Identical sequences are highlighted by *blue* letters. The filled *red* box highlights the two-residue acidic motif at the start of the N-terminal α-helix, followed by a conserved α-helical region (open box). The two cysteine residues unique to ANGPTL4 are shown in *yellow* boxes. *Panel* **B** shows the time-dependent unfolding of 10 µM LPL by 1 µM ANGPTL3 (*green circles*) as defined by the appearance of bimodality in the isotope envelopes for peptide 131–165. For comparison the unfolding of LPL by 1 µM ANGPTL4$^{1-159}$ is shown as solid (wild-type) and hatched (E15K) *gray* lines. Spontaneous unfolding is shown by *black* triangles. *Panel* **C** shows the residual lipolytic activity of 10 µM LPL incubated for 10 min alone or in the presence of 1 µM ANGPTL3, 1 µM ANGPTL3 *and* 30 µM GPIHBP1, or 1 µM ANGPTL4$^{1-159}$.

spaces *by locally produced ANGPTL4* is important for regulating the amount of active LPL that actually reaches the surface of capillaries, thereby serving to match the efficiency of intravascular lipolysis to the metabolic requirements of nearby parenchymal cells.

Finally, we propose that the resistance of GPIHBP1-bound LPL to ANGPTL4- and ANGPTL3-mediated inhibition serves to focus catalytically active LPL at the luminal surface of capillary endothelial cells. It is interesting, that human population studies have not uncovered a correlation between plasma ANGPTL4 levels and triglyceride concentrations (*Robciuc et al., 2010*; *Smart-Halajko et al., 2010*) implying that the regulation of LPL activity in this compartment is far more convoluted. The low-to-moderate concentrations of circulating ANGPTL4 and ANGPTL3 in the blood are obviously insufficient to inhibit LPL in the presence of triglyceride-rich lipoproteins (*Nilsson et al., 2012*). With that said, it is important to acknowledge that elevated plasma levels of ANGPTL4 protein achieved by transgenic overexpression or systemic administration clearly breached this protective barrier as these mice have increased plasma triglyceride levels and vastly reduced post-heparin LPL activity levels (*Köster et al., 2005*; *Lichtenstein et al., 2007*; *Nilsson et al., 2012*). Similarly, the protection yielded by GPIHBP1-binding may also not be completely impregnable. With concentrations predicted to saturate LPL with GPIHBP1 (*Figure 2A* and *Figure 3B*), we did observe a minor increase in LPL unfolding/inactivation in the presence of large amounts of ANGPTL4 in vitro. Intravascular lipolysis is thus a highly tunable biological process where the net activity is critically dependent on a complex interplay between local concentrations of cognate activators and inhibitors and their partitioning between different compartments. Using selective inhibitor repression to focalize a given

biological activity is not uncommon; for example, during fibrinolysis plasmin-bound to its substrate (polymerized fibrin) escapes inhibition by $\alpha_2$-antiplasmin (*Lijnen, 2001*).

Genome-wide association studies have shown that the common polymorphic variant ANGPT-L4[E15K] (often denoted E40K) is linked to mild hypotriglyceridemia and reduced risk of coronary artery disease (*Dewey et al., 2016*; *Helgadottir et al., 2016*; *Stitziel et al., 2016*; *Yin et al., 2009*). While the impact of ANGPTL4[E15K] on lipid metabolism in humans is clear, the structure–function relationship(s) responsible for its reduced biological activity have been unclear. Our studies combined with previously published studies, suggest that the Glu-to-Lys replacement in ANGPTL4 affects two properties. First, the E15K polymorphism reduces the ability of ANGPTL4 to inhibit LPL in vitro—to the degree that ANGPTL4[E15K] functions no better than ANGPTL3 (*Shan et al., 2009*). With HDX-MS, we found that ANGPTL4[E15K] and ANGPTL3 catalyze LPL unfolding to a similar degree (*Figure 6*). Second, although the E15K polymorphism has no impact on protein synthesis, secretion, oligomerization, or secretion, it still limits extracellular accumulation of the coiled-coil domain of ANGPTL4 in cultured cell systems as well as in adenovirus-transfected mice (*Yin et al., 2009*). It has been hypothesized that the effects of the E15K polymorphism on plasma triglyceride levels relate to a short half-life of the ANGPTL4[E15K] oligomer (*Yin et al., 2009*). In the current study, we provide evidence supporting that idea—by showing that the Glu-to-Lys substitution in ANGPTL4 causes protein destabilization *via* cooperative unfolding of the first $\alpha$-helix in ANGPTL4′s coiled-coil domain (*Figure 6*). Negative charges at the N-terminus of $\alpha$-helices generally stabilize coiled-coil domains *via* electrostatic interactions with the $\alpha$-helix dipole moment (*Johnson et al., 2014*; *Kohn et al., 1997*). Accordingly, the charge-reversal in ANGPTL4[E15K] would be expected to lead to protein destabilization—in agreement with our HDX-MS data (*Figure 5*). We suspect that this mechanism is a contributing factor to the lower half-life of ANGPTL4[E15K].

By HDX-MS, we found that ANGPTL3 catalyzed the inactivation of LPL, although markedly less efficient compared with ANGPTL4 (*Figure 6*). GPIHBP1-binding also protected LPL from inactivation by ANGPTL3, suggesting that the mechanism that we proposed for ANGPTL4-mediated LPL inactivation also applies to ANGPTL3. Of note, ANGPTL3, ANGPTL4, and ANGPTL8 all share a two-residue acidic motif in their first $\alpha$-helix along with a highly conserved GLLQGLG motif (*Figure 6A*). Intervention studies with mAbs recognizing this region in ANGPTL3 and ANGPTL4 reduce plasma triglyceride levels in animal models, emphasizing the functional importance of this region (*Desai et al., 2007*; *Gusarova et al., 2015*). In light of these observations, we suspect that the function of other ANGPTL proteins would be sensitive to missense mutations that introduce charge reversals and compromise conformational stability. Thus far, however, mutations comparable to the E15K substitution in ANGPTL4 have not been reported in ANGPTL3 or ANGPTL8.

## Materials and methods

### Purified proteins and reagents

A recombinant secreted version of human GPIHBP1 was produced in *Drosophila* S2-cells (Invitrogen, CA) as a fusion protein with domain 3 (D3) of human uPAR (*Gårdsvoll et al., 2007*). For practical reasons, the recombinant GPIHBP1 was mutated within its N-terminal disordered region (Arg[38]→Gly); that amino acid substitution provides a higher purification yield without altering LPL-binding properties (*Mysling et al., 2016*). Bovine LPL was purified from fresh bovine milk by heparin-Sepharose, hydroxyapatite, and Superdex HR200 size-exclusion chromatography (*Cheng et al., 1985*). The coiled-coil domain of wild-type human ANGPTL4 (ANGPTL[wt]) and ANGPTL4[E15K] (residues 1–159 with an N-terminal methionine and a C-terminal 6×His-tag) was produced in *E. coli* BL21 (DE3) with a pet29a vector (*Robal et al., 2012*). Full-length ANGPTL3 produced in HEK293 cells was purchased from ProSpec (Ness-Ziona, Israel). A synthetic peptide corresponding to residues 1–33 of human GPIHBP1 was obtained at a purity of >95% from TAG-Copenhagen A/S (Copenhagen, Denmark).

For HDX-MS experiments, purified ANGPTL4[1–159] preparations were dialyzed for 18 hr at 4°C against 10 mM formic acid (pH 4.1) using 2000 MWCO Slide-A-Lyzer dialysis cassettes (Thermo Scientific). Protein concentrations were determined by absorbance at 280 nm.

## ANGPTL4-mediated unfolding of LPL assessed by pulse-labeling HDX-MS

To determine the rates of spontaneous and ANGPTL4-catalyzed LPL unfolding, we incubated LPL alone or in the presence of ANGPTL4. In brief, 10 µM bovine LPL was incubated in 8.8 mM $Na_2HPO_4$, 0.7 mM HCOOH, 150 mM NaCl (pH 6.9) at 25°C with 300 RPM mixing, either alone or in the presence of different amounts of ANGPTL4$^{1-159/wt}$ (2 µM, 1 µM, 0.5 µM, 0.25 µM) or ANGPTL4$^{1-159/E15K}$ (2 µM, 1 µM, 0.5 µM). Identical volumes were added to ensure identical buffer compositions with different concentrations of ANGPTL4. Solutions were incubated for 0.5, 2.5, 5, 10, 15, or 30 min. The folding status of LPL was probed by adding $D_2O$-buffer (10 mM $Na_2HPO_4$, 150 mM NaCl, pD 7.4) to 70% (v/v) $D_2O$ and allowing hydrogen–deuterium exchange to occur for 10 s before quenching the exchange reaction by adding 1 vol ice-cold quench buffer (100 mM $Na_2HPO_4$, 0.8 M tris-(2-carboxyethyl) phosphine (TCEP), 2 M urea in $H_2O$, pH 2.5). Quenched samples were incubated on ice for 2 min to reduce disulfide bonds and were subsequently snap-frozen in liquid $N_2$ and stored at –80°C until analysis by MS. Labeling of all individual time points was replicated 2 to 4 times. The degree of unfolding of LPL's hydrolase domain was estimated by quantifying the bimodal isotope distribution for the 'diagnostic' peptic peptide 131–165, which contains two residues of LPL´s catalytic triad. This was accomplished by fitting the bimodal isotope envelope to two Gaussian distributions and comparing the areas of the resultant fits.

## Alkylation of cysteines in ANGPTL4$^{wt}$

To reduce disulfide bonds and alkylate the free thiols in ANGPTL4$^{1-159}$, the pH was adjusted to 6.5 by adding 3 volumes of 100 mM Bis-Tris, pH 6.5. After adding TCEP (10 mM final concentration), the reduction of disulfide bonds proceeded for 1 hr at room temperature. Subsequently, thiols were alkylated by adding 0.1 vol of 0.5 M iodoacetamide followed by a 4 hr incubation in the dark at room temperature. Alkylated samples were dialyzed overnight in 10 mM formic acid (pH 4.1) to adjust the pH and remove excess TCEP and iodoacetamide.

## Impact of GPIHBP1 on ANGPTL4-catalyzed LPL-unfolding

To study the effect of purified GPIHBP1 on the unfolding of LPL by ANGPTL4$^{1-159}$, we first formed LPL•GPIHBP1 complexes by incubating 11 µM LPL with 33 µM GPIHBP1$^{1-131}$, 33 µM GPIHBP1$^{34-131}$, 33 µM GPIHBP1$^{1-33}$, or a mixture of 33 µM GPIHBP1$^{34-131}$ and 33 µM GPIHBP1$^{1-33}$ for 2 min at 25°C. LPL (11 µM) without ligands was incubated in parallel. Unfolding was initiated by adding 10% (v/v) 20 µM ANGPTL4$^{1-159}$ and allowing unfolding to proceed for 10 min in 10 mM $Na_2HPO_4$, 0.1 mM HCOOH, and 150 mM NaCl (pH 7.1) at 25°C. Similar data were recorded for spontaneous unfolding of 10 µM LPL, except that GPIHBP1 concentrations were reduced to 10 µM and the incubation time was increased to 45 min. Unfolding of LPL´s hydrolase domain was assessed with pulse-labeling in 70% $D_2O$ for 10 s, followed by quenching and reduction on ice for 2 min. Labeling of each sample was replicated between 3 and 6 times.

## Determination of lipase activity

Before measuring lipase activity with Intralipid containing [$^3$H]triolein, we took care to replicate the conditions used for the HDX-MS experiments (i.e., 10 µM LPL at 25°C, identical amounts of ANGPTL4 and GPIHBP1, and identical buffer conditions). The ANGPTL4-induced unfolding of LPL was quenched by 10-fold dilution in ice-cold 20 mM Tris (pH 8.5) containing 5 mM deoxycholic acid (DOC) and 0.1 mM sodium dodecyl sulfate (SDS) (DOC/SDS buffer). Lipase activities were determined by adding 5 µl of LPL (diluted 100-fold in DOC/ SDS buffer) to 195 µl of incubation mixtures containing Intralipid with incorporated [$^3$H]triolein and with heat-inactivated rat serum as a source of the activator apolipoprotein C-II (*Larsson et al., 2013*).

In some experiments, the protective effects of GPIHBP1 on lipase activity, were assessed in 96 well plates where 60 µl of 15 nM LPL in PBS buffer (pH 7.4) containing 0.01% (v/v) of Trition-X100 were pre-incubated in the presence or absence of 7.5 nM ANGPTL4 and 300 nM GPIHBP1 at 21°C and shaking at 600 rpm. To determine lipase activity at a given pre-incubation time, 90 µl incubation mixture containing Intralipid and human serum (as a source of apolipoprotein C-II) were added and the incubation was continued for 25 min at room temperature. After terminating the reaction by adding Triton X-100 to a final concentration of 2.5% (v/v), the amount of lipase-generated, non-

esterified fatty acids (NEFA) were quantified using a NEFA-HR(2) assay according to the manufacturer's protocol (Wako Diagnostics). Measurements were performed in triplicates.

## Protein dynamics of ANGPTL4 measured by HDX-MS

Labeling with deuterium was initiated by 3.3-fold dilutions of 10 μM ANGPTL4$^{1–159/wt}$ or ANGPTL4$^{1–159/E15K}$ (in 10 mM Na$_2$HPO$_4$, 150 mM NaCl) into the equivalent deuterated buffer (pD 7.4), yielding the following exchange conditions: 70% (v/v) D$_2$O, pH 7.1 (as measured with identical solutions prepared from protiated reagents). Solvent exchange was allowed to proceed for 10, 100 or 1000 s at 25°C with 300 RPM mixing. At the indicated time intervals, the exchange reaction was quenched with 1 vol ice-cold quench buffer and protein disulfides reduced for 2 min on ice. On-line digestion with pepsin yielded 41 peptides shared by ANGPTL4$^{wt}$ and ANGPTL4$^{E15K}$, providing 96% sequence coverage. Full deuteration controls were prepared by allowing LPL samples to exchange in the same buffer for 48 or 72 hr at 37°C. No additional deuterium uptake could be observed after 48 hr.

## MS analysis of HDX-labeled samples

Deuterium incorporations into proteins was determined after on-line pepsin digestion of the samples with an HDX-modified reversed-phased chromatographic system coupled to a Synapt G2 electrospray ionization mass spectrometer (Waters, Milford, MA) (*Mysling et al., 2016*).

## Acknowledgements

We acknowledge Gry Ellis Rasmussen and Seungwon Jung for expert technical assistance and John Post for artwork. *This work was supported by a Leducq Transatlantic Network grant (12CVD04), NIH grants HL090553 and HL087228, Rigshospitalets Forskningsudvalg (KKK), the Swedish Research Council for Medicine and Health (2015–02942), the Swedish Heart and Lung Foundation (20130684) and the Wenner-Gren Foundations. GO and ML are shareholders in Lipigon Pharmaceuticals AB where GO serves as a board member. The other authors have no financial interests to declare.

## Additional information

### Competing interests

SGY: Reviewing editor, *eLife*. ML: shareholder in Lipigon Pharmaceuticals AB. GO: shareholder and board member in Lipigon Pharmaceuticals AB. The other authors declare that no competing interests exist.

### Funding

| Funder | Grant reference number | Author |
| --- | --- | --- |
| Leducq Transatlantic Network | 12CVD04 | Stephen G Young<br>Michael Ploug |
| National Institutes of Health | HL090553 | Stephen G Young |
| National Institutes of Health | HL087228 | Stephen G Young |
| Rigshospitalet | 123 | Kristian Kølby Kristensen |
| Swedish Research Council for Medicine and Health | 2015-02942 | Gunilla Olivecrona |
| Swedish Heart and Lung Foundation | 20130684 | Gunilla Olivecrona |

The funders had no role in study design, data collection and interpretation, or the decision to submit the work for publication.

### Author contributions

SM, KKK, ML, TJDJ, Acquisition of data, Analysis and interpretation of data, Drafting or revising the article; OK, Acquisition of data, Analysis and interpretation of data, Drafting or revising the article, Contributed unpublished essential data or reagents; AB, GO, SGY, Analysis and interpretation of

data, Drafting or revising the article, Contributed unpublished essential data or reagents; MP, Conception and design, Analysis and interpretation of data, Drafting or revising the article

Author ORCIDs

Michael Ploug, http://orcid.org/0000-0003-2215-4265

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
