## [Decision Letter]

Thank you for submitting your article "ANGPTL4 catalyzes unfolding of the hydrolase domain in lipoprotein lipase and that unfolding is counteracted by GPIHBP1" for consideration by *eLife*. Your article has been favorably evaluated by Harry Dietz (Senior Editor) and three reviewers, one of whom is a member of our Board of Reviewing Editors. The following individuals involved in review of your submission have agreed to reveal their identity: Tobias C Walther (Reviewer #2); Joseph Witztum (Reviewer #3).

The reviewers have discussed the reviews with one another and the Reviewing Editor has drafted this decision to help you prepare a revised submission.

Summary:

The manuscript by Mysling et al. reports an extension of their recent work in *eLife* on the regulation of structure and function of LPL. They again use MS analysis of hydrogen-deuterium exchange to follow time-dependent changes in LPL structure and correlate these changes with assays of lipase activity. They demonstrate that sub-stoichiometric amounts of ANGPTL4 accelerate unfolding of the serine hydrolase domain, but not lipid binding domain of LPL, and correspondingly reduce its enzymatic activity. The human ANGPTL4 E15K variant that is associated with reduced triglyceride levels is less active in catalyzing LPL unfolding. ANGPTL3 also catalyzes unfolding of the LPL serine hydrolase domain, but is less active than ANGPTL4. The effects of ANGPTL4 on LPL unfolding and catalytic activity are counteracted by the endothelial cell transporter, GPIHBP1. Collectively, these results provide a biochemical explanation for the restriction of LPL activity to the right place and an explanation for reduced activity of the ANGPTL4 variant. Overall there was a high degree of enthusiasm for the manuscript from the three reviewers, but there is one issue that needs to be addressed experimentally and several other points requiring further discussion.

Essential Revisions:

1) The authors describe an experimental system in which LPL unfolding is irreversible. However, there is data from Saskia Nehr's group that ANGPTL4 inhibits LPL in a reversible fashion. The studies are not directly comparable, but it certainly raises the issue as to whether irreversibility of LPL unfolding in these studies is a specific feature of the experimental conditions used. This question should be further addressed experimentally.

2) Since GPIHBP1 binding to LPL stabilizes it and presumably protects it from inactivation by ANGPTL4 and ANGPTL3 – why then when heparin is given and LPL is released into the fluid phase plasma is it not totally inactivated by the combinations of ANGPTL3 and 4? Would it not be as in Figure 1 in solution and away from GPIHBP1? And how does for example, apoC-III, fit in here? This seems a major paradox. Does the heparin somehow change the LPL into a dimer that is then resistant to the ANGPTL proteins? Or is it all a matter of balance (stoichiometry) and numbers of molecules activating/inhibiting etc.? These questions should be discussed.

3) It was recently demonstrated that ANGPTL4 mediated intracellular degradation of LPL in adipocytes. Do the authors think this is related? E.g. Could ANGPTL4 mediated unfolding lead to a degradation pathway? Would the described mutant therefore lead to enhanced adipose tissue release of functional LPL in the affected individual? These questions should be discussed.

4) In vivo, LPL bound to GPIHBP1 interacts with a very large TRL and in the context of that binding, there is also bound apoC-II in the mix, and the different ANGPTL3/4/8 not to mention apoC-III. Can the authors discuss how they envision the described mechanism to play out in the context of LPL mediated hydrolysis of TG in a TRL?

---

## [Author Response]

*[…] Essential Revisions:*

1) The authors describe an experimental system in which LPL unfolding is irreversible. However, there is data from Saskia Nehr's group that ANGPTL4 inhibits LPL in a reversible fashion. The studies are not directly comparable, but it certainly raises the issue as to whether irreversibility of LPL unfolding in these studies is a specific feature of the experimental conditions used. This question should be further addressed experimentally.

We are very happy for this opportunity to further elaborate on this topic as the apparent inconsistency in the literature as mentioned by the reviewer has persisted for several years. We actually consider our data on the sub-stoichiometric inhibition of LPL with ANGPTL4 presented in the original manuscript as a very strong argument in favor of the catalytic inhibition of LPL via an *irreversible* unfolding mechanism, during which each event leaves LPL permanently inactivated enabling the subsequent release of ANGPTL4, which can engage in the next unfolding event. Nonetheless, it is also true that the conditions defined by the HDX-MS conditions are unique in the sense that they use relatively high concentrations of LPL (10 µM). To interrogate the irreversibility of the LPL inhibition by ANGPTL4 with more traditionally used conditions, we therefore repeated the founding experiment leading Lafferty et al. 2013 to conclude that ANGPTL4 acts via a reversible non-competitive mechanism. We also discuss why we think they might have been misled by these experiments. These data are now included in the manuscript as Figure 1—figure supplement 4 and is referred to in the Discussion section, second paragraph. In brief; we were able to partially reproduce their inhibition data with ANGPTL4 inhibition recorded at 4°C in the presence of 1 mM DOC. However, if we omit DOC – a well-known LPL stabilizing anionic-detergent – from the reaction mixture, we observe an efficient time-dependent inhibition of LPL by sub-stoichiometric amounts of ANGPTL4 even at 4°C – in agreement with an irreversible inactivation mechanism.

Further inspired by the reviewer comments and our new protection data with GPIHBP1, we decided to design an additional experiment addressing this important question using more physiological relevant agents. This new information is now added to the manuscript as Figure 3 and discussed in the Results section:

“To further interrogate the mode of action of ANGPTL4-catalyzed LPL inhibition, we added 300 nM intact GPIHBP1 [12-fold above the K_D_ for the GPIHBP1•LPL interaction (Mysling et al., 2016)] to LPL (15 nM) that had been partially inactivated – either by incubating the enzyme at 21°C (spontaneous inactivation) or by incubating the enzyme with 7.5 nM ANGPTL4 (catalyzed inactivation). […] If ANGPTL4-mediated inactivation of LPL were to be reversible, we would have expected to find reactivation of LPL during the incubation with GPIHBP1.”

*2) Since GPIHBP1 binding to LPL stabilizes it and presumably protects it from inactivation by ANGPTL4 and ANGPTL3 – why then when heparin is given and LPL is released into the fluid phase plasma is it not totally inactivated by the combinations of ANGPTL3 and 4? Would it not be as in Figure 1 in solution and away from GPIHBP1? And how does for example, apoC-III, fit in here? This seems a major paradox. Does the heparin somehow change the LPL into a dimer that is then resistant to the ANGPTL proteins? Or is it all a matter of balance (stoichiometry) and numbers of molecules activating/inhibiting etc.? These questions should be discussed.*

This is a highly relevant objection and opens the discussion of the highly convoluted regulation of lipid metabolism in vivo. In the hindsight, we did perhaps in the Discussion focus too narrowly on the impact of the GPIHBP1 – ANGPTL4-axis on LPL regulation as this was at the epicenter of what was addressed experimentally in the manuscript. What we originally emphasized was the importance and impact of this interaction in focusing the assembly of a functional lipolysis platform on the capillary endothelial surface – rather than addressing what occurs in the blood circulation. This is self-evidently a reductionist point-of-view as one compartment cannot be evaluated without the other. We have therefore added this element to the Discussion:

“Finally, we propose that the resistance of GPIHBP1-bound LPL to ANGPTL4- and ANGPTL3-mediated inhibition serves to focus catalytically active LPL at the luminal surface of capillary endothelial cells. […] With concentrations predicted to saturate LPL with GPIHBP1, we did observe a minor increase in LPL unfolding/inactivation in the presence of large amounts of ANGPTL4 in vitro(Figure 2 and Figure 3).”

*3) It was recently demonstrated that ANGPTL4 mediated intracellular degradation of LPL in adipocytes. Do the authors think this is related? E.g. Could ANGPTL4 mediated unfolding lead to a degradation pathway? Would the described mutant therefore lead to enhanced adipose tissue release of functional LPL in the affected individual? These questions should be discussed.*

The recent data from Sander Kerstens laboratory on a possible role of ANGPTL4 in intracellular degradation of LPL is indeed interesting and we realized that we have not discussed the potential association of the ANGPTL4-catalysed unfolding of LPL to this report in sufficient detail. We have therefore expanded this section considerably and included some possible links to lysosomal degradation:

“First, in the secretory pathway of adipocytes, where GPIHBP1 is absent, we suspect that LPL would be more susceptible to ANGPTL4-mediated unfolding. […] Such a mechanism might account for the increased LPL secretion from *Angptl4*^-/-^ adipocytes compared to wild-type adipocytes (Dijk et al., 2016). In keeping with the latter idea, we speculate that human adipocytes that express ANGPTL4 E15K polymorphism would secrete elevated levels of LPL.”

*4)* In vivo*, LPL bound to GPIHBP1 interacts with a very large TRL and in the context of that binding, there is also bound apoC-II in the mix, and the different ANGPTL3/4/8 not to mention apoC-III. Can the authors discuss how they envision the described mechanism to play out in the context of LPL mediated hydrolysis of TG in a TRL?*

Once again this is a very relevant, important and highly complex question. However, we consider it almost impossible to provide a reasonable answer – without oversimplifying the subject – as the microenvironment entailing the marginated TRL on the capillary endothelial surface is highly complex. The physiochemical properties of this interface depends on so many variables – biochemical as well as biological – that it is a daunting task even to define which of the reactions rates are limiting. To mention a few complications: the surface-bound nature of most reactants confounds a reasonable calculation of their active concentrations; the change in size and surface tension of the TRL during lipolysis needs to be taken into account; the membrane dynamics and clustering of GPIHBP1 in lipid rafts are likely to influence the number of recruited LPL-GPIHBP1 complexes to the TRLs; and even the sheer force exerted by the blood flow on the marginated TRL is likely a contributing factor. We would therefore prefer not to enter a detailed description of the regulation of this interesting and highly complex lipolysis platform as we run the risk of presenting a model flawed by too many assumptions and too little actual data. In the revised manuscript, we have tried to emphasize this complexity by stating:

“Intravascular lipolysis is thus a highly tunable biological process where the net activity is critically dependent on a complex interplay between local concentrations of cognate activators and inhibitors and their partitioning between different compartments.”